# CLEX: Continuous Length Extrapolation for Large Language Models

**Guanzheng Chen**[1,2,3,*] **Xin Li**[2,3,†] **Zaiqiao Meng**[4] **Shangsong Liang**[1,5,†] **Lidong Bing**[2,3]

[1]Sun Yat-sen University [2]DAMO Academy, Alibaba Group
[3]Hupan Lab, 310023, Hangzhou, China [4]University of Glasgow
[5]Mohamed bin Zayed University of Artificial Intelligence
`guanzzh.chen@gmail.com, {xinting.lx,l.bing}@alibaba-inc.com`
`zaiqiao.meng@glasgow.ac.uk, liangshangsong@gmail.com`

## Abstract

Transformer-based Large Language Models (LLMs) are pioneering advances in many natural language processing tasks, however, their exceptional capabilities are restricted within the preset context window of Transformer. Position Embedding (PE) scaling methods, while effective in extending the context window to a specific length, demonstrate either notable limitations in their extrapolation abilities or sacrificing partial performance within the context window. Length extrapolation methods, although theoretically capable of extending the context window beyond the training sequence length, often underperform in practical long-context applications. To address these challenges, we propose **C**ontinuous **L**ength **EX**trapolation (**CLEX**) for LLMs. We generalise the PE scaling approaches to model the continuous dynamics by ordinary differential equations over the length scaling factor, thereby overcoming the constraints of current PE scaling methods designed for specific lengths. Moreover, by extending the dynamics to desired context lengths beyond the training sequence length, CLEX facilitates the length extrapolation with impressive performance in practical tasks. We demonstrate that CLEX can be seamlessly incorporated into LLMs equipped with Rotary Position Embedding, such as LLaMA and GPT-NeoX, with negligible impact on training and inference latency. Experimental results reveal that CLEX can effectively extend the context window to over $4\times$ or almost $8\times$ training length, with no deterioration in performance. Furthermore, when evaluated on the practical LongBench benchmark, our model trained on a 4k length exhibits competitive performance against state-of-the-art open-source models trained on context lengths up to 32k. Our code is available at `https://github.com/DAMO-NLP-SG/CLEX`.

## 1 Introduction

Transformer-based large language models (LLMs), such as GPT-4 (OpenAI, 2023) and LLaMA (Touvron et al., 2023a;b), have now emerged as the state-of-the-art models in various natural language processing (NLP) tasks. However, these models grapple with the limitations inherent to the Transformer architecture - mainly, a preset context window, beyond which performance plummets catastrophically (Press et al., 2022). The quadratic complexity of the attention mechanism renders training LLMs with a larger context window extraordinarily resource-intensive. Prior works (Dai et al., 2019; Beltagy et al., 2020; Bulatov et al., 2022) have proposed circumventing full context length access via hierarchical architecture or sparse attention, albeit at the expense of forfeiting partial context information.

Recently, there have been two lines of methods aimed at efficiently extending the pre-trained context length of LLMs, both centred on position embedding (PE). The first line of methods, known as *PE scaling*, are proposed to effectively extend the context window of LLMs integrated with Rotary

---

*This work was done during the internship of Guanzheng Chen at Alibaba DAMO Academy.
†Corresponding authors.

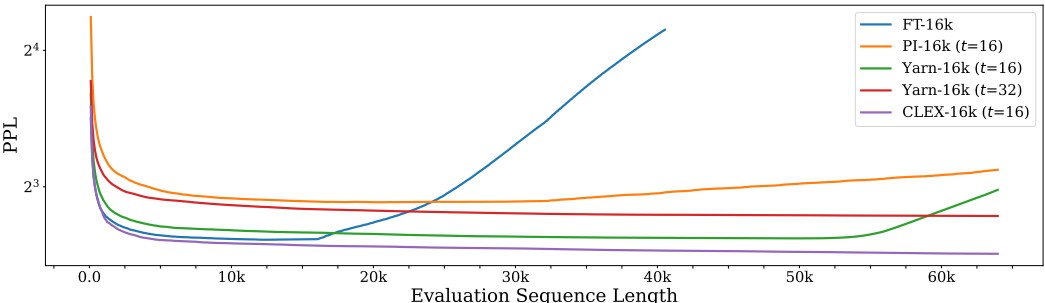

Figure 1: The PPLs of our CLEX and various baselines tested on 64k context length.

Position Embedding (RoPE) (Su et al., 2022). They allow LLMs to access longer context by scaling either position indices (Chen et al., 2023) or frequency basis (Rozière et al., 2023; Peng et al., 2023) of RoPE, demonstrating remarkable performance in long-context applications. However, such methods are designed for extending the context length corresponding to a fixed scaling factor, which either restricts their ability to extrapolate to longer sequences (when using small factors) or impairs the performance even within the native context window (when using large factors) as shown in Figure 1. On the other hand, *length extrapolation* methods (Press et al., 2022; Sun et al., 2023; Chi et al., 2022; 2023), typified by ALiBi (Press et al., 2022), strive to achieve test-time context length extension (i.e., "training on short, testing on long") by substituting position embeddings with additional biases, where the biases encode positional information to the attention scores. Despite their impressive capability in language modelling, ALiBi-like methods usually struggle in the practical tasks requiring long-context dependency (Pal et al., 2023) (also see §4.3).

In this work, we present **C**ontinuous **L**ength **EX**trapolation (**CLEX**), a novel approach that efficiently extrapolates the context window of LLMs through continuous PE scaling. Concretely, we propose a unified view of PE scaling via generalising the PE scaling methods to the transition of frequency basis. Upon it, we formulate the PE scaling as a *continuous dynamical system*, which models the transition of frequency basis through the continuous dynamics over the length scaling factor. We argue that previous PE scaling methods, training models using fixed (discrete) scaling factors, overlook the progressively continuous dynamics over the gradually length-extending process. This ensnares themselves in the aforementioned dilemma between extrapolating the length and preserving the performance within shorter lengths. In contrast, our CLEX exploits a neural ordinary differential equation (ODE) (Chen et al., 2018), parameterised by an up-and-down projection layer with lightweight parameters to learn these continuous dynamics, enabling fine-grained extending to long context. More essentially, by extending the dynamics beyond training length, CLEX empowers models to progressively extrapolate to longer contexts even when trained with short sequences.

CLEX can serve as a drop-in component for RoPE-based LLMs, such as LLaMA (Touvron et al., 2023a;b) and GPT-NeoX (Black et al., 2022), with negligible overhead in computation and parameters size. We evaluate the performance of CLEX on two datasets: (1) a subset of RedPajama-Book (Computer, 2023) for long-context language modelling, and (2) LongBench (Bai et al., 2023) for long-context practical tasks. Empirically, CLEX demonstrates remarkable length extrapolation ability in language modelling, which can extend the context window to more than $4\times$ training length without any performance deterioration. For example, LLaMA-2-7B trained with CLEX on 16k context length achieves comparable perplexities when testing on 16k and 64k tokens, respectively. By scaling the base model scale from 7B to 13B, CLEX exhibits an expanded extrapolation scope from $4\times$ to almost $8\times$ training length. To be complementary, we also conduct instruction tuning (Wei et al., 2022) with the proposed CLEX on the sequences of 4k length. The resulting model, when evaluated on the LongBench benchmark, is on par with current state-of-the-art open-source models trained on context lengths up to 32k. These findings underscore the effectiveness of CLEX in extrapolating context length, signifying its efficiency for developing long-context LLMs.

## 2 PRELIMINARIES

### 2.1 ROTARY POSITION EMBEDDING (ROPE)

Rotary Position Embedding (RoPE) (Su et al., 2022) has recently emerged as the most prevailing positional encoding method in open-source LLMs like LLaMA. It integrates both absolute and relative positional information for Transformer models. Given a position index $m \in [1, L]$, RoPE injects the absolute positional information into $\boldsymbol{x} \in \mathbb{R}^d$ via the transformation $f \colon \mathbb{R}^d \to \mathbb{R}^d$ as:

$$f(\boldsymbol{x}, m, \boldsymbol{\theta}) = \boldsymbol{R}_{\boldsymbol{\theta},m}\boldsymbol{x}, \tag{1}$$

where $\boldsymbol{\theta} \in \mathbb{R}^{\lfloor d/2 \rfloor}$ is the rotation *frequency basis* and $\theta_i = 10,000^{-2i/d}$; $\boldsymbol{R}_{\boldsymbol{\theta},m} \in \mathbb{R}^{d \times d}$ is a block diagonal matrix formed by the elements

$$(\boldsymbol{R}_{\boldsymbol{\theta},m})_i = \begin{bmatrix} \cos m\theta_i & -\sin m\theta_i \\ \sin m\theta_i & \cos m\theta_i \end{bmatrix}, \text{ for } i = 1, 2, ..., \lfloor d/2 \rfloor. \tag{2}$$

The transformation in Eq. (1) is applied to the query and key vectors during self-attention. When calculating the attention score for the query vector $\boldsymbol{q}_m \in \mathbb{R}^d$ at position $m$ and the key vector $\boldsymbol{k}_n \in \mathbb{R}^d$ at position $n$, we have

$$(\boldsymbol{R}_{\boldsymbol{\theta},m}\boldsymbol{q}_m)^\top (\boldsymbol{R}_{\boldsymbol{\theta},n}\boldsymbol{k}_n) = \boldsymbol{q}_m \boldsymbol{R}_{\boldsymbol{\theta},n-m}\boldsymbol{k}_n. \tag{3}$$

Hence, the relative positional information $\boldsymbol{R}_{\boldsymbol{\theta},n-m}$ is implicitly incorporated into the attention scores. However, even given the relative information, LLMs trained with RoPE, e.g., LLaMA, still cannot achieve reasonable performance beyond the pre-trained context length.

### 2.2 PE SCALING METHODS

To extend the context length $L$, several strategies are proposed to adjust the position embedding by scaling either the position index $m$ or frequency basis $\boldsymbol{\theta}$ in Eq. (1). Formally, we define $t = L'/L$ as the length scaling factor where $L'$ denotes the desired extended length. While Chen et al. (2023) introduces scaling the index $m$ by *Position Interpolation (PI)* as

$$f_t^{\text{PI}}(\boldsymbol{x}, m, \boldsymbol{\theta}) = f(\boldsymbol{x}, \frac{m}{t}, \boldsymbol{\theta}). \tag{4}$$

This strategy maintains the position indices within the range $[1, L]$, while effectively extending the processed range to $[1, t \cdot L]$ by minimal fine-tuning steps on $t \cdot L$ sequences. On the other hand, Peng et al. (2023) proposes Yarn, a.k.a. NTK-Aware Scaled RoPE, extends the context window by frequency basis scaling (FBS). This strategy is similarly utilised by CodeLLaMA (Rozière et al., 2023). Formally, the FBS methods are denoted as

$$f_t^{\text{FBS}}(\boldsymbol{x}, m, \boldsymbol{\theta}) = f(\boldsymbol{x}, m, \boldsymbol{\theta}_t), \tag{5}$$

where $\boldsymbol{\theta}_t$ is the scaled frequency basis. Specifically, $\theta_{t,i} = \theta_i \cdot (t)^{-2i/(d-2)}$ in Yarn and $\theta_{t,i} = \theta_i \cdot 100^{-2i/d}$ in CodeLLaMA.

## 3 METHODOLOGY

This section demonstrates the details of CLEX. We first generalise the PE scaling to a continuous dynamical system in a unified manner (see §3.1). On top of the continuous dynamical system, CLEX employs the neural ODE, parameterised by an up-and-down projection layer, to adaptively learn the continuous dynamics during PE scaling (see §3.2). In §3.3, we introduce the training strategy of CLEX that distributes the continuous dynamics beyond the training sequence length, thereby enabling the generalisation of continuous PE scaling to achieve the length extrapolation.

### 3.1 POSITION EMBEDDING SCALING: A UNIFIED VIEW

Given the various methods that extend models' context length through position indices scaling and frequency basis scaling, we first show that the transformations applied to position indices are essentially casting the frequency basis, which is formalised in Theorem 1.

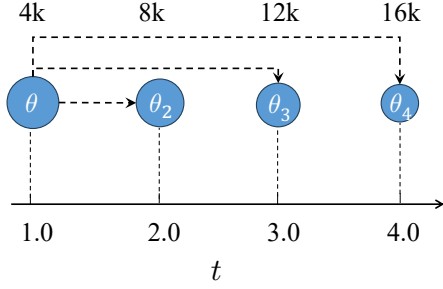 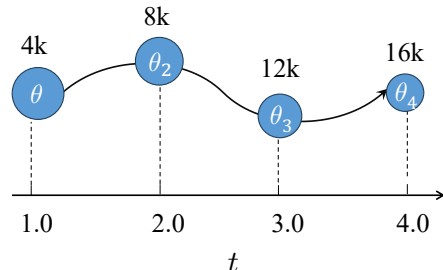

Figure 2: The graphical model of discrete PE scaling (left) and our continuous PE scaling (right).

**Theorem 1.** *For the transformation $\mathcal{T}$ to position index $m$, there exists an equivalent transformation $\boldsymbol{\mathcal{T}}$ to frequency basis $\boldsymbol{\theta}$ in Eq. (1), namely*

$$f(\boldsymbol{x}, \mathcal{T} \cdot m, \boldsymbol{\theta}) = f(\boldsymbol{x}, m, \boldsymbol{\mathcal{T}} \odot \boldsymbol{\theta}), \tag{6}$$

*where $\boldsymbol{\mathcal{T}} = [\mathcal{T}]_{i=1}^{d/2}$ and $\odot$ denotes the element-wise transformation.*

**Proof.** *From Eq. (1), we have $f(\boldsymbol{x}, \mathcal{T} \cdot m, \boldsymbol{\theta}) = \boldsymbol{R}_{\boldsymbol{\theta}, \mathcal{T}m} \boldsymbol{x}$ and $f(\boldsymbol{x}, m, \boldsymbol{\mathcal{T}} \odot \boldsymbol{\theta}) = \boldsymbol{R}_{\boldsymbol{\mathcal{T}} \odot \boldsymbol{\theta}, m} \boldsymbol{x}$. For any $\boldsymbol{\mathcal{T}} = [\mathcal{T}]_{i=1}^{d/2}$,*

$$(\boldsymbol{R}_{\boldsymbol{\theta}, \mathcal{T}m})_i = \begin{bmatrix} \cos \mathcal{T}m\theta_i & -\sin \mathcal{T}m\theta_i \\ \sin \mathcal{T}m\theta_i & \cos \mathcal{T}m\theta_i \end{bmatrix} = \begin{bmatrix} \cos m(\boldsymbol{\mathcal{T}} \odot \theta_i) & -\sin m(\boldsymbol{\mathcal{T}} \odot \theta_i) \\ \sin m(\boldsymbol{\mathcal{T}} \odot \theta_i) & \cos m(\boldsymbol{\mathcal{T}} \odot \theta_i) \end{bmatrix} = (\boldsymbol{R}_{\boldsymbol{\mathcal{T}} \odot \boldsymbol{\theta}, m})_i. \tag{7}$$

Hence, there is a unified form for PE scaling that consistently projects the frequency basis by $\boldsymbol{\alpha}(t)$:

$$f_t(\boldsymbol{x}, m, \boldsymbol{\theta}) = f\left(\boldsymbol{x}, m, \boldsymbol{\alpha}(t) \odot \boldsymbol{\theta}\right), \tag{8}$$

where $\boldsymbol{\alpha}(t)$ is a single-variable transformation defined over the length scaling factor $t$. Through this unified formulation, PI (Chen et al., 2023) and Yarn (Peng et al., 2023) can be viewed as the special cases when plugging $\boldsymbol{\alpha}^{\text{PI}}(t) = [1/t]_{i=1}^{d/2}$ and $\boldsymbol{\alpha}^{\text{Yarn}}(t) = \left[t^{-2i/(d-2)}\right]_{i=1}^{d/2}$ into Eq. 8, respectively.

Note that $\boldsymbol{\theta}_t = \boldsymbol{\alpha}(t) \odot \boldsymbol{\theta}$ denotes the scaled frequency basis at context length of $t \cdot L$ and $\boldsymbol{\theta}_1 = \boldsymbol{\theta}$ (namely $\boldsymbol{\alpha}(1) = 1$). As illustrated in Figure 2, this indicates a progressive chain across discrete $t$ values that

$$\boldsymbol{z}(t) = \boldsymbol{z}(1) + \log \boldsymbol{\alpha}(t) = \boldsymbol{z}(t-1) + \log \frac{\boldsymbol{\alpha}(t)}{\boldsymbol{\alpha}(t-1)}, \tag{9}$$

where $\boldsymbol{z}(t) = \log \boldsymbol{\theta}_t$.

By continuizing the progressive chain, we can formulate the PE scaling as a continuous dynamical system, with the continuous dynamics of frequency basis $d\boldsymbol{z}(t)/dt$ as

$$\frac{d\boldsymbol{z}(t)}{dt} = \frac{d \log \boldsymbol{\alpha}(t)}{dt}. \tag{10}$$

In essence, recent PE scaling methods, concentrating on manually formulating the $\boldsymbol{\alpha}(t)$, are equivalent to applying various dynamics for frequency basis that enable models to adapt to longer contexts.

## 3.2 CONTINUOUS PE SCALING VIA NEURAL ODE

Even given the continuous dynamics of frequency basis, previous methods are inherently designed for extending the context length at discrete $t$ values. For example, PI (Chen et al., 2023) fine-tunes the model on a specific scaling factor $t$ to extend the context window length to $t \cdot L$. One potential issue of these methods, as depicted in Figure 1, is that they are susceptible to overfitting to the specified frequency basis, leading to either poor extrapolation ability to longer lengths beyond training or performance drops within short lengths, or both in some cases. Therefore, our CLEX aims to build a *continuous* PE scaling, which induces the model to adapt the frequency basis corresponding to a continuous scope of $t$ as illustrated in Figure 2 (right).

Recall that previous PE scaling, corresponding to a manually defined $\boldsymbol{\alpha}(t)$, implies the constant dynamics in Eq. (10). In our method, we utilise a variable function $g \colon \mathbb{R}^{d/2} \to \mathbb{R}^{d/2}$ to model the dynamics, hence towards a more general and flexible view as:

$$\frac{d\boldsymbol{z}(t)}{dt} = g(\boldsymbol{z}(t), t). \tag{11}$$

By restricting the function to be associated with the latent states $\boldsymbol{z}(t)$, $g$ is capable of capturing the fine-grained changes of frequency basis during the length-extending process. However, it is non-trivial to manually define the $\boldsymbol{z}(t)$-aware function $g$. Here, we directly parameterise the function using the neural network $\boldsymbol{\phi}$. Therefore, for any $t' \in [1, t]$, there is a neural ODE modelling the scaling of frequency basis as

$$\boldsymbol{z}(t') = \boldsymbol{z}(1) + \int_{1}^{t'} g_{\boldsymbol{\phi}}(\boldsymbol{z}(t), t) dt, \tag{12}$$

where the frequency basis at the length $t' \cdot L$ can be derived by $\boldsymbol{\theta}_{t'} = \exp(\boldsymbol{z}(t'))$.

More specifically, we adopt an up-and-down projection as the neural network, expressed as:

$$g_{\boldsymbol{\phi}}(\boldsymbol{z}(t), t) = \boldsymbol{W}_{\text{down}} \cdot \sigma\left(\boldsymbol{W}_{\text{up}} \cdot \boldsymbol{z}(t)\right) + \xi_t, \tag{13}$$

where $\boldsymbol{W}_{\text{up}} \in \mathbb{R}^{\frac{d}{2} \times \lambda d}$ and $\boldsymbol{W}_{\text{down}} \in \mathbb{R}^{\lambda d \times \frac{d}{2}}$ are the transformation matrices, of which the parameters are determined by the amplification factor $\lambda$; $\sigma$ is the SiLU activation function and $\xi_t$ is the scalar embedding typifying the scaling procedure at factor of $t$. Here, we adopt the constant dynamics of Yarn as the $\xi_t$ for speeding up convergence, namely

$$\xi_t = \frac{d \log \boldsymbol{\alpha}^{\text{Yarn}}(t)}{dt} = -\left[\frac{2i}{(d-2) \cdot t}\right]_{i=1}^{d/2} \tag{14}$$

### 3.3 CONTINUOUS LENGTH EXTRAPOLATION: TRAIN ON SHORT, TEST ON LONG

Continuous PE scaling can serve as a more adaptive and flexible PE scaling method to extend the context length to a given training length $L^{\text{Train}}$. Unlike the previous PE scaling methods built on a larger scaling factor, which would lead to inferior performance on the lengths corresponding to smaller counterparts, the continuous PE scaling would enable non-destructively generalisation to larger scaling factors via adaptive continuous dynamics. Therefore, by simply extending the continuous dynamics beyond the factor $t = L^{\text{Train}}/L$ during training (where we denote the desired scaling factor as $t^{\text{Train}}$), we can access the *continuous length extrapolation* (CLEX) method, which achieves the capability of "training on short, testing on long".

Moreover, to learn the neural ODE in Eq. (12) for continuous $t$, we randomly sample $t' \in [1, t^{\text{Train}}]$ for each training step, enabling the model to adapt to the broad scope frequency basis without overfitting a specific one. Note that the frequency basis is bound with the position index in Eq. (1). This reveals the aforementioned training involves inconsistency between the frequency basis and position indices: the frequency basis is varied corresponding to the $t' \in [1, t^{\text{Train}}]$, while the position indices are fixed as $\{1, 2, \ldots, L^{\text{Train}}\}$. Here, we propose the *position extrapolation* strategy to address this consistency. Contrary to PI, which shrinks the position indices into the context length, we enlarge the position indices $\{1, 2, \ldots, L^{\text{Train}}\}$ of the trained sequences up to the range $[1, t' \cdot L]$ for each training step. The position indices can be acquired by uniformly scaling to $\{1 \cdot s, 2 \cdot s, \ldots, L^{\text{Train}} \cdot s\}$ where $s = t' \cdot L / L^{\text{Train}}$, or alternatively, by randomly sampling $L^{\text{Train}}$ of indices from $[1, t' \cdot L]$. Empirically, we found that random sampling generally performs better. More discussions can be found in §4.2.

During inference, the ideal scenario is to acquire the frequency basis corresponding to each sequence length. However, this approach is computationally demanding. To improve efficiency, we first cache some frequency basis derived from $g_{\boldsymbol{\phi}}$ for $K$ discrete $t$ values as $\{t_k | k \in [1, K]\}$. For each sequence with a length of $L^{\text{Infer}}$ during inference, we employ the frequency basis corresponding to the nearest upper bound within $t_k \cdot L$ for $k = 1, \ldots, K$. Through this, our method introduces negligible time cost compared to naive inference of LLMs.

## 4 EXPERIMENTS

In this section, we conduct a thorough evaluation of CLEX's performance in terms of handling long contexts and its extrapolation capabilities. We compare our approach against other methods covering

| Methods | Train Length | Evaluation Length | | | | | | | | | |
|---|---|---|---|---|---|---|---|---|---|---|---|
| | | 4096 (4k) | | 8192 (8k) | | 16,384 (16k) | | 32,768 (32k) | | 65,536 (64k) | |
| | | PPL | ACC. | PPL | ACC. | PPL | ACC. | PPL | ACC. | PPL | ACC. |
| LLaMA-2 | 4k | 6.04 | 58.18 | 20.54 | 44.50 | >100 | 22.43 | >1000 | 12.70 | >1000 | 10.64 |
| CodeLLaMA | 16k | 7.60 | 54.88 | 7.40 | 55.19 | 7.33 | 55.30 | 15.12 | 44.70 | 52.02 | 31.16 |
| Naive FT | 16k | 5.98 | 58.83 | 5.93 | 58.91 | 5.91 | 58.58 | 18.31 | 43.04 | >100 | 26.05 |
| PI | 16k | 5.90 | 59.05 | 5.71 | 59.44 | 5.72 | 59.87 | 6.05 | 58.5 | 8.75 | 52.02 |
| Yarn ($t$=16) | 16k | 6.50 | 57.28 | 5.71 | 59.57 | 5.73 | 59.87 | 5.99 | 58.13 | 8.51 | 52.62 |
| Yarn ($t$=32) | 16k | 6.61 | 57.12 | 5.94 | 58.27 | 5.96 | 58.04 | 6.08 | 57.73 | 6.22 | 57.98 |
| CL-Scaling | 16k | 24.99 | 37.84 | 5.86 | 59.08 | 5.87 | 59.05 | 10.56 | 50.47 | 41.09 | 34.16 |
| ALiBi | 4k | 6.34 | 58.01 | 6.39 | 57.8 | 6.41 | 57.78 | 6.50 | 57.47 | 6.51 | 56.44 |
| RandomPos | 4k | 5.88 | 58.49 | >100 | 34.23 | >1000 | 18.27 | >1000 | 9.31 | >1000 | 7.44 |
| | 4k | **5.86** | **59.21** | 5.70 | 59.62 | 5.87 | 58.93 | 14.53 | 47.55 | 30.51 | 35.33 |
| CLEX | 8k | 5.98 | 58.75 | 5.78 | 59.44 | 5.71 | 59.64 | 5.99 | 58.66 | 11.74 | 47.50 |
| | 16k | 5.88 | 59.21 | **5.68** | **59.73** | **5.52** | **60.28** | **5.55** | **60.06** | **5.64** | **59.94** |

Table 1: Perplexity (PPL) and next-token-prediction accuracy (ACC.) on language modeling with evaluation lengths from 4k to 64k. We train the LLaMA-2-7B using length extrapolation methods on 4k length and PE scaling methods on 16k length, while reporting the results of CLEX trained across 4k, 8k and 16k. CL-Scaling denotes training LLaMA-2-7B with the scaling method of CodeLLaMA but using our training data. The training loss curves are depicted in Figure 9.

both length extrapolation (i.e., ALiBi (Press et al., 2022) and random positions (denoted as Random-Pos) (Ruoss et al., 2023)) and PE scaling methods (i.e., PI (Chen et al., 2023) and Yarn (Peng et al., 2023)). We primarily conduct experiments on the LLaMA-2-7B model. For the language modelling, we train our model and the baselines on 2B tokens extracted from Redpajama-Book (Computer, 2023), which is collected from Pile-Books3 (Gao et al., 2020) and PG-19 (Rae et al., 2019) datasets. The performance of the models is assessed based on perplexity and next-token-prediction accuracy, with evaluation sequence lengths up to 64k. Furthermore, we conduct instruction tuning for LLaMA-2-7B using CLEX on the UltraChat dataset (Ding et al., 2023b). The evaluation is performed on the LongBench benchmark (Bai et al., 2023), where we compare our model with GPT-3.5-turbo and other LLaMA-2-based open-source models designed for handling long context. Further details about baselines and training configuration will be discussed in Appx. §A, as well as more experimental results and ablations in Appx. §B.

## 4.1 LONG-CONTEXT LANGUAGE MODELLING

**CLEX achieves length extrapolation.** We first report the experimental results of baselines and CLEX on language modelling, with the evaluation length from 4k to 64k. As shown in Table 1, our CLEX consistently demonstrates remarkable performance in length extrapolation, being able to extrapolate the context length to more than $4\times$ training length without any performance drops. Taking CLEX-4k as an example, its PPL on 4k sequence (training length) is comparable to that on 16k sequence (5.86 vs. 5.87). When evaluated on the sequences within 16k, CLEX-4k is on par with or even better than all of the compared methods trained on lengths up to 16k. Moreover, with the increase in training length, our CLEX not only exhibits promising generalisation capability to very long contexts (up to 64k) but also guarantees performance on short sequences.

We also found that discrete PE scaling methods (i.e., PI and Yarn) have *self-extending* property: training with scaled frequency basis equips the model with the ability to extrapolate to further-scaled counterparts (see Appx. §B.2 for more discussions.). As depicted in Figure 1, however, the extrapolation capability of these methods is limited, accompanied by a significant performance decline even within the naive context length. This indicates the inherent challenge of achieving a delicate balance between extrapolation to longer lengths and performance maintenance within short lengths when using the discrete scaling factor. In contrast, CLEX tackles this issue via learnable continuous dynamics, providing a more fine-grained extrapolation while preserving the performance for the internal context.

Note that ALiBi may extend further than CLEX trained on 4k sequences (though typically producing inferior results), our experiments reveal that these gains may come at the cost of long-term information, leading to underperformance in long-context practical tasks (see §4.3 for more details).

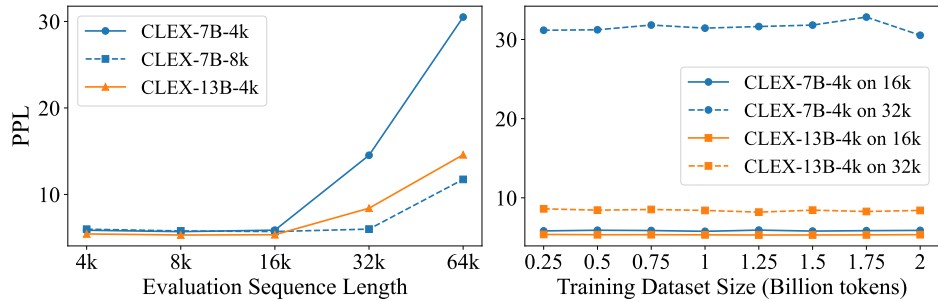

Figure 3: Left: The PPLs of CLEX on different evaluation sequence lengths with 7B and 13B parameter sizes. Right: The PPLs of CLEX cross variable training data size with different parameter sizes and evaluation lengths.

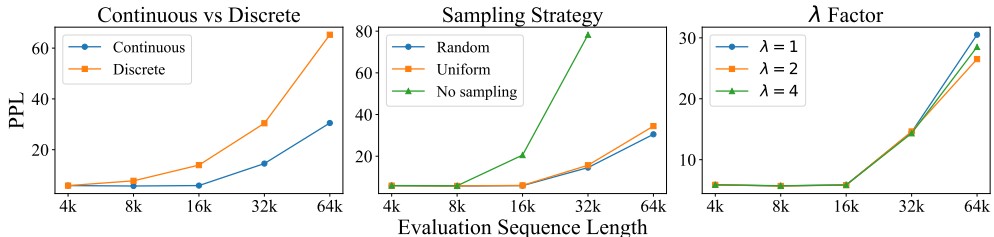

Figure 4: The ablation studies for continuous dynamics, sampling strategies and $\lambda$ factor in Eq. (13).

**The scaling law for the extrapolation ability of CLEX.**    To investigate the effectiveness of CLEX over the scale of the base model and training data size, we further port our method to LLaMA-2-13B. As depicted in Figure 3, when trivially extending the base model scale from 7B to 13B, our CLEX demonstrates an increased capacity to extrapolate to longer context lengths. Specifically, the extrapolation ability of CLEX-13B trained on 4k length approaches that of CLEX-7B trained on 8k. While the training data scale, more surprisingly, does not significantly impact the extrapolation capability of CLEX. Models trained with 0.25B or 2B tokens with 4k sequence length achieve comparable PPLs when evaluating on 16k or 32k lengths in Figure 3, indicating the negligible margins from the larger training data size. This also implies that CLEX can efficiently extend the context length of LLMs through minimal training steps resembling PI and Yarn.

Based on these findings, we propose a scaling law for CLEX: to scale the context length of LLMs to moderately desired lengths (e.g., 16k → 64k), one should proportionally enlarge the training sequence lengths (e.g., 4k → 16k). For scaling the context length up to considerably long lengths (e.g., >200k), the parameter size of the base model should be correspondingly increased while maintaining the training length, since the contexts may take more footprints than model parameters. Note that scaling the training data does not directly affect the extrapolation ability of CLEX, but may be implicitly incorporated when scaling the base pre-trained LLMs.

## 4.2  ABLATION STUDY

We now conduct three types of ablations to investigate the efficacy of the components in CLEX:

**Continuous dynamics.**    To learn the continuous dynamics using neural ODE, we adopt a distinct training approach that involves sampling the scaling factor $t$ for each data batch. Here we seek to explore if the exceptional extrapolation ability of CLEX is solely derived from the variable $t$ rather than the continuous dynamics. We employ the discrete Yarn method with $t = 16$, that undergoes the same training procedure of CLEX but removes the ODE parameters, serving as a discrete baseline. In Figure 4 (left), we discover that the discrete approach equipped with the random-sampled $t$ significantly underperforms our CLEX, indicating the essentiality of the learnable continuous dynamics in CLEX for accessing the extrapolation ability.

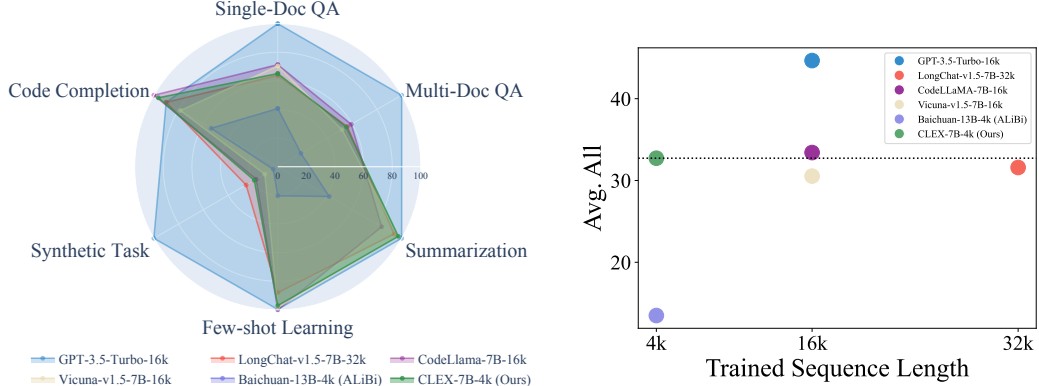

Figure 5: Left: the average scores for each domain of tasks in LongBench. Right: the average scores of all tasks corresponding to the training length of each model. Note that CLEX is trained on 4k sequence length but directly tested on 16k context length without truncation.

**Position extrapolation.** We adopt the position extrapolation strategy, which extends the scope of position indices in training sequences by sampling from a broader range, to reconcile the inconsistency between frequency basis and position indices during the training process. In this study, we examine the impact of various sampling strategies (uniform or random) and contrast them with the naive position indices. The results in Figure 4 underscore the efficacy of position extrapolation in CLEX, without which the extrapolation ability of models declines significantly. Furthermore, random sampling slightly performs better than uniform sampling, so we adopt it across all experiments.

**The parameter scale of ODE.** We also study the impact of parameter size of the neural ODE in CLEX. The parameter size is determined by the $\lambda$, namely the amplification factor in Eq. (13). In Figure 4, we plot the results of CLEX with $\lambda = 1, 2, 4$, where they achieve similar performance. Note that the parameter size of neural ODE in CLEX is quite small even when $\lambda = 4$, as the dimension $d$ in Eq. (13) is usually equal to 128. Although it is possible to enhance CLEX with larger $\lambda$ (e.g., 32), we set the $\lambda=1$ in all experiments for the minimal effect on inference latency.

## 4.3 Evaluation on Long-Context Benchmark

To ascertain the comprehensive performance of CLEX in real-world scenarios, we further conduct an evaluation on the zero-shot LongBench benchmark. This benchmark encompasses a broad range of tasks, such as question-answering, summarization, and code completion, where the evaluation length ranges from 5k to 15k. We perform a pilot instruction tuning for LLaMA-2-7B by employing CLEX on the UltraChat dataset, with a sequence length of 4k. During inference, we harness all models to tackle the context length of 16k, thereby ensuring the comprehensive exploitation of contextual information in the tasks. As depicted in Figure 5, we present the average scores of each domain in LongBench for CLEX, in comparison to the GPT-3.5-Turbo-16k model and strong open-source LLMs like LongChat-v1.5-7B-32k and CodeLLaMA-7B-16k.

Generally, when trained with sequences of 4k length, CLEX holds its own against any open-source LLMs that are trained on lengths up to 32k. In the specific domains of Summarization, Few-shot Learning, and Code Completion, CLEX on LLaMA-2-7B remains competitive with or even surpasses the GPT-3.5-Turbo-16k. We note that the Baichuan-13B-4k, pre-trained with ALiBi (Press et al., 2022), demonstrates marked underperformance on the LongBench although with a larger parameter size. Additionally, similar poor results are achieved by ALiBi when applying it upon LLaMA-2-7B using the same training procedure as CLEX (see Appx. §B.5). This could likely be attributed to ALiBi's overemphasis on local context through the attention bias, which, while advantageous for language modelling, restricts access to long-context information in practical tasks. In contrast, CLEX directly extends the context length of LLMs without imposing any constraints on context, which consistently achieves superior extrapolation ability on both language modelling and the LongBench. This substantiates the considerable potential of CLEX to serve as the state-of-the-art approach for extrapolating the context length of LLMs to excel in long-context applications.

In addition, we highlight that our CLEX merely introduces minuscule inference latency. Given a context length of 16k in LongBench with a generation length of 512, the generation throughput between our CLEX-7B and LLaMA-2-7B is comparable (27.8 tokens/s vs 28.3 tokens/s, in a single A100), when using the cache mechanism introduced in §3.3.

## 5 RELATED WORK

**Hierarchical Architecture / Sparse Attention.**   To overcome the quadratic complexity of attention, Dai et al. (2019) proposes the Transformer-XL that handles the long sequence at segment level by Transformer, with these segments interacting through a recurrence mechanism. The Recurrent Memory Transformer (Bulatov et al., 2022) refines this mechanism by incorporating special memory tokens into the recurrence, which is capable of scaling the context length to the millions (Bulatov et al., 2023). On the other hand, Child et al. (2019); Beltagy et al. (2020) proposed using the sparse attention to circumvent the full access to the long sequences, hence reducing the complexity. The sparse attention has been adopted by Ding et al. (2023a) to scale the context length of transformers into the billions. However, these methods sacrifice the utilisation of the entire sequence during attention, resulting in an inevitable loss of some contextual information. Additionally, modifications to the model architecture make these methods challenging to apply to existing pre-trained LLMs. Conversely, our CLEX serves as a drop-in component for LLMs, can efficiently extend the capacity of models to tack the entire long sequences without explicit drops of context information.

**Length Extrapolation.**   Building on the foundation laid by ALiBi (Press et al., 2022), a series of works (Sun et al., 2023; Chi et al., 2022; 2023) seek to train the Transformer-based models on a short length, while directly testing on longer counterparts. These methods substitute the position embedding with bias introduced into attention scores, thereby incorporating positional information. Notably, the bias typically gives higher profits to closer tokens. This mechanism intuitively amplifies the local context for each token at the expense of distant information. Consequently, these length-extrapolation methods encounter challenges in effectively handling long contexts in practical applications (Pal et al., 2023). However, our CLEX demonstrates remarkable effectiveness in practical tasks such as summarization, indicating the de facto extrapolation ability for applications.

**Position Embedding (PE) Scaling.**   Recent research has sought to extend the context length of Transformers through the scaling of the extensively employed RoPE. Specifically, Chen et al. (2023) proposed position interpolation, a method that efficiently extends the context window by scaling the position index within RoPE. In a similar vein, Peng et al. (2023); Rozière et al. (2023) opted to scale the frequency basis, achieving superior performance. However, these methods necessitate training (or fine-tuning) on the desired extended length. As a result, they exhibit a limited ability to extrapolate beyond the trained length and even suffer from performance drops within the shorter lengths. In CLEX, we generalise the discrete PE scaling to a continuous counterpart, hence uniformly extrapolating the context length of LLMs while preserving the performance within short lengths.

## 6 CONCLUSION

We have presented the Continuous Length EXtrapolation (CLEX), a novel approach that efficiently extrapolates the context length of Large Language Models (LLMs) to over 4x the training (fine-tuning) length without any decline in performance. CLEX utilises the neural ODE to learn the continuous dynamics over the length scaling factor during PE scaling, hence enabling fine-grained extension for the frequency basis in the RoPE. We conduct thorough experiments to investigate the effectiveness of CLEX compared to a variety of strong LLMs, covering the language modelling task and LongBench benchmark. The experimental results have demonstrated the exceptional extrapolation ability of CLEX, where our CLEX trained with a sequence length of 4k holds the potential to remain competitive to any open-source long-context LLMs (e.g., CodeLLaMA) trained on lengths up to 32k. These results highlight the potential of CLEX as a state-of-the-art approach for efficiently extrapolating the context length of LLMs, paving the way for advancements in long-context applications. By scaling the base model size up, we found CLEX can be correspondingly enhanced and subsequently is capable of extrapolating the model to a longer context length. This property indicates the tempting effectiveness of CLEX in the era of LLMs.

ACKNOWLEDGEMENTS

This work was substantially supported by DAMO Academy through DAMO Academy Research Intern Program. Shangsong Liang was supported by the National Natural Science Foundation of China (Grant No. 61906219) and the Mohamed bin Zayed University of Artificial Intelligence, United Arab Emirates.

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

# A  EXPERIMENTAL DETAILS

## A.1  BASELINES

We compare our CLEX with a variety of baselines in language modelling, all of which are trained with the subset of 2B tokens from Pile-Books3 and PG-19 except LLaMA-2 and CodeLLaMA that are evaluated by the open-source pre-trained checkpoints. We train the PE scaling and length extrapolation methods with 16k and 4k sequence lengths, respectively. Specifically, we use the scaling factor $t = 4$ to train models using 16k sequence length for PI following Chen et al. (2023). While for Yarn, we train models with the scaling factor of 16 and 32 (but also using the 16k sequence length), respectively, following the settings in Peng et al. (2023). For the ALiBi, we directly remove the RoPE in LLaMA-2 and train models with the attention bias. For the random positions, we sample the position indices from [1, 16k] while train models with 4k sequence length. We also utilise the PE scaling method from CodeLLaMA to train LLaMA-2-7B using our training data, which is denoted as CL-Scaling.

## A.2  TRAINING DETAILS

We use a subset from the Redpajama-Book (Computer, 2023) as the training data for language modelling, which is collected from Pile-Books3 and PG-19 datasets and the subset comprises approximately 2B tokens. We set the learning rate of 2e-5 for all models, which are optimised by Adam optimiser (Kingma & Ba, 2015). The batch size is set to 64k tokens for 7B models and 16k tokens for 13B models. The maximum desired $t$ during training in CLEX (namely $t^{\text{Train}}$ in §3.3) is set as 16 for LLaMA-2. We utilise the Flash Attention (Dao et al., 2022; Dao, 2023) to support efficient training. The amplification factor of ODE layer $\lambda$ is set as 1 for all 7B models and 2 for 13B models. For the instruction tuning, we train our model using the UltraChat dataset for 1 epoch, starting with the checkpoint after the training of language modelling. The training procedure of CLEX is shown in Alg. 1.

---

**Algorithm 1** Training Procedure of CLEX

1: **repeat**
2:     Given a batch of sequences $\mathcal{B}$ with length of $L^{\text{Train}}$;
3:     sample $t' \sim [1, t^{\text{Train}}]$;
4:     calculate $\boldsymbol{z}(t')$ by Eq. (12);
5:     $\boldsymbol{\theta}_{t'} = \exp(\boldsymbol{z}(t'))$;
6:     sample (randomly or uniformly) $t' \cdot L^{\text{Train}}$ position indices from $[1, t^{\text{Train}} \cdot L^{\text{Train}}]$;
7:     calculate RoPE with $\boldsymbol{\theta}_{t'}$ and sampled position indices;
8:     train the model with RoPE on $\mathcal{B}$ with language modelling objective.
9: **until** converged

---

## A.3  EVALUATION DETAILS

For language modelling, we evaluate all models with a similar subset to the training set but containing 20 million tokens. For different evaluation lengths, we group the test set into corresponding lengths of sequences. We found fixing the scaling factor as the training one would hamper the general performance and extrapolation ability of baselines, therefore, we proportionally enlarge the scaling factor when the evaluation length is beyond the training length. For example, when evaluating on 32k length, we set the scaling factor of PI as 8 rather than the training factor of 4. We also adopt the log scaling trick[1] for all baselines and our model, where we scale the attention scores by the $\max\{1, \log_{L^{\text{Train}}} L^{\text{Test}}\}$. We found it can improve the performance when evaluating on lengths beyond training length as shown in Figure 8. For the LongBench, we follow the setups from Bai et al. (2023), where the decoding method is set to greedy search. We set the $t_k$ in §3.3 as $\{1, 2, 3, 4\}$, to cache the frequency basis corresponding to $\{4k, 8k, 12k, 16k\}$.

---

[1] https://spaces.ac.cn/archives/8823

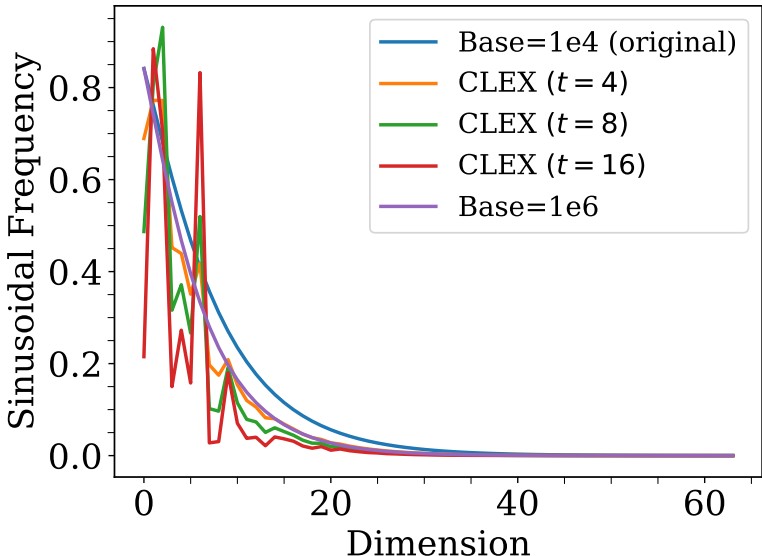

Figure 6: The sinusoidal frequency over the dimension of the frequency basis. The lower dimension denotes the higher frequency.

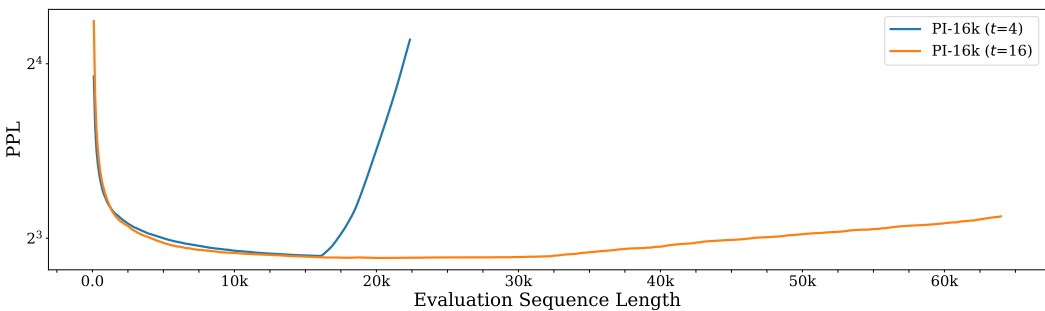

Figure 7: The results of PI-16k (trained with $t = 4$) on 64k sequence length, while evaluating with $t = 4$ and $t = 16$.

## B  ADDITIONAL RESULTS

### B.1  VISUALISATION OF LEARNED FREQUENCY BASIS

In Figure 6, we show the frequency basis according to different sequence lengths after the training of CLEX on LLaMA-2 with a 4k sequence length. The plot reveals that CLEX tends to upscale the high frequencies at certain dimensions while simultaneously downscaling some others. More surprisingly, we have observed that the frequency bases associated with different $t$ values in CLEX exhibit an isotropic behaviour, that the dimensions where downscaling and upscaling occur are similar across different $t$ values, with larger $t$ values resulting in further scaling. This finding may contribute to the design of heuristics PE scaling methods without training.

### B.2  THE SELF-EXTENDING PROPERTY OF PE SCALING METHODS

Previous PE scaling methods usually evaluate models with a specified scaling factor during training. We found this would significantly hamper the performance evaluation beyond the training length. Hence, we adjust the scaling factor of the previous PE scaling method corresponding to the evaluation lengths. In Figure 7, PI-16k (trained with $t = 4$) achieves non-trivial results when evaluating

| Models | Train Length | Evaluation Length | | | |
|---|---|---|---|---|---|
| | | 32K | 64K | 128K | 256K |
| CLEX-Phi-2 | 32K | 5.11 | 5.17 | 6.55 | >10 |
| CLEX-Mixtral-8x7B | 32K | 2.56 | 2.53 | 2.57 | 3.78 |

Table 2: Perplexity (PPL) of CLEX-Phi-2 and CLEX-Mixtral-8x7B on language modelling with evaluation lengths from 32k to 256k.

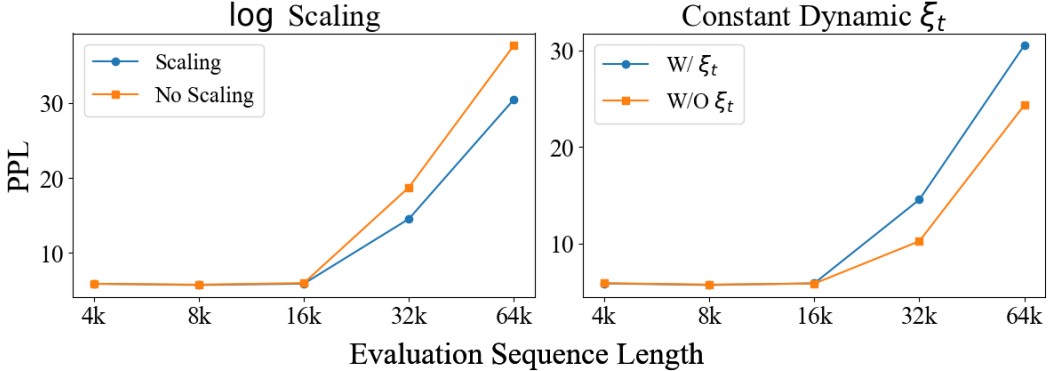

Figure 8: The ablation of $\log$ scaling (**left**) and constant dynamic $\xi_t$ (**right**) for CLEX trained on 4k sequence length.

with $t = 16$ on 64k length, which significantly outperforms the naive evaluation using $t = 4$. Note that Yarn is naively trained with larger $t$, so it is not necessary to adjust the scaling factor for Yarn during evaluation. When the evaluation lengths become longer, however, we believe the self-extending property would benefit the Yarn as well. Regarding these, the extrapolation ability from the self-extend in previous PE scaling methods is still far weaker than CLEX as illustrated in Figure 1.

### B.3 RESULTS OF MORE MODELS

We also extend CLEX to more models beyond LLaMA-2, i.e., Phi-2[2] and Mixtral-8x7B (Jiang et al., 2024). The results in Table 2 further support the general effectiveness of CLEX, where the context lengths are extrapolated to 4x∼8x training length across various LLMs.

### B.4 ABLATION ON SCALAR EMBEDDING

The learnable dynamic in Eq. (13) is based on the constant dynamic $\xi_t$ from Yarn. To conduct an ablation on the impact of $\xi_t$, we remove $\xi_t$ in Eq. (13) and train the LLaMA-2-7B on 4k sequence length that undergoes the same training procedure. From Figure 8, we found the training of CLEX without $\xi_t$ would not hamper the extrapolation ability (or even better). However, as depicted in Figure 10, training with $\xi_t$ can speed up the convergence and produce a more stable loss curve. This is essential for training larger LLMs on longer sequence lengths, so we integrate the $\xi_t$ into the learnable dynamics.

### B.5 RESULTS OF LONGBENCH

We list the numerical results of our CLEX trained on 4k compared with a variety of baselines trained on up to 32k, covering single-document question answering (Table 3), multi-document question answering (Table 4), summarization (Table 5), few-shot learning (Table 6), synthetic (Table 7), and

---

[2]https://huggingface.co/microsoft/phi-2

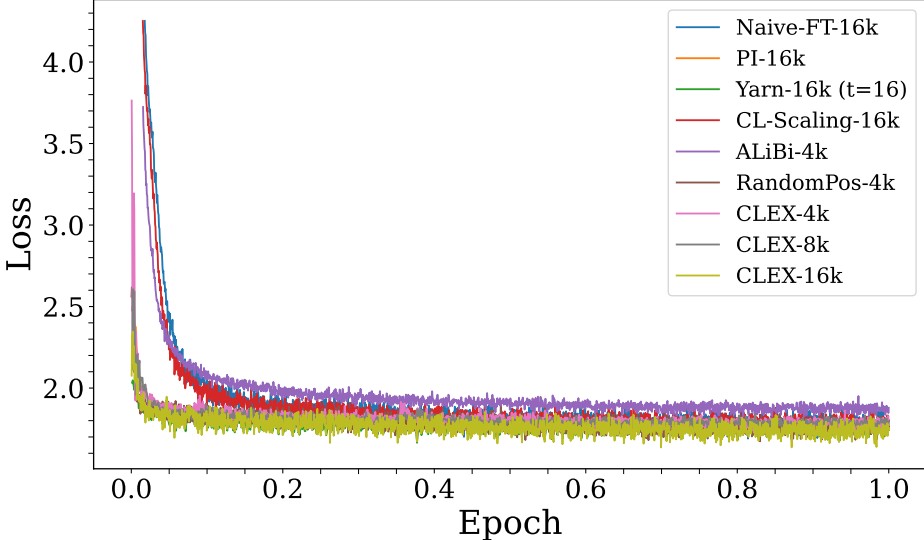

Figure 9: The training loss curves of various baselines and our CLEX models in Table 1. We train all models upon LLaMA-2-7B on 2B tokens for one epoch.

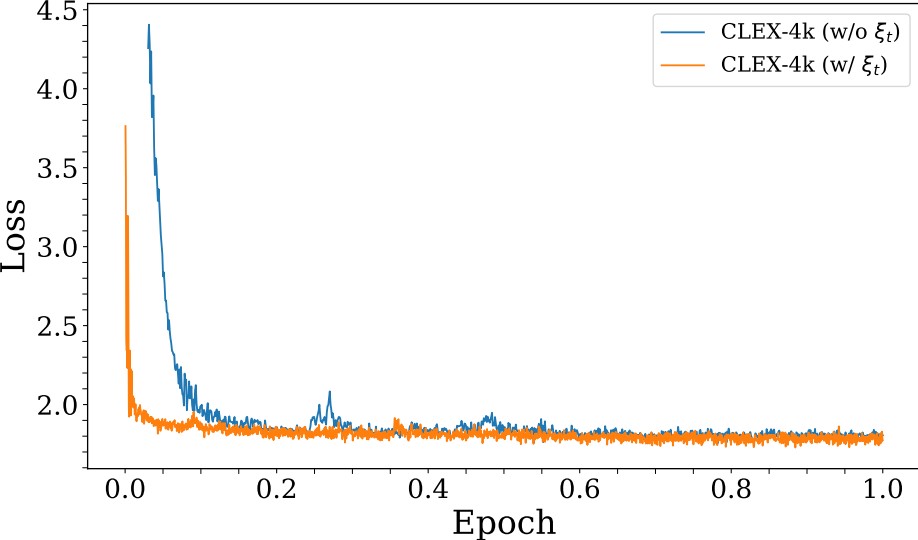

Figure 10: The training loss curves of our CLEX-4k trained on LLaMA-2-7B with or without the constant dynamics ($\xi_t$) in Eq. (14).

code completion (Table 8) tasks. Note that the average sequence length of most tasks in Long-Bench ranges from 5k to 15k, so the samples would be truncated into the sequence length within the supported context length (if necessary) for baselines, except Baichuan-13B-4k, ALiBi-7B-4k and CLEX-7B-4k which enables length extrapolation are evaluated with a context window of 16k. Baichuan-13B-4k is a model pre-trained with ALiBi on a sequence length of 4k, while ALiBi-7B-4k is the model trained with ALiBi that undergoes the same training procedure as our CLEX. We found that both ALiBi-based models significantly underperform in the LongBench benchmark when evaluated with sequence length (16k) beyond their training length, indicating the challenges of their extrapolation ability in practical tasks. Our CLEX, however, achieves well-performing results across most tasks even trained with 4k but tested with 16k length, which further reveals the superior length extrapolation ability of our method.

| Models | NarrativeQA | Qasper | MultiFieldQA-en | MultiFieldQA-zh | AVG. |
|---|---|---|---|---|---|
| GPT-3.5-Turbo-16k | 23.6 | 43.3 | 52.3 | 61.2 | 45.1 |
| Llama2-7B-chat-4k | 18.7 | 19.2 | 36.8 | 11.9 | 21.65 |
| LongChat-v1.5-7B-32k | 16.9 | 27.7 | 41.4 | 29.1 | 28.78 |
| CodeLLaMA-7B-16k | 22.93 | 30.69 | 43.37 | 31.76 | 32.19 |
| XGen-7B-8k | 18 | 18.1 | 37.7 | 14.8 | 22.15 |
| InternLM-7B-8k | 12.1 | 16.7 | 23.4 | 33.6 | 21.45 |
| Vicuna-v1.5-7B-16k | 19.4 | 26.1 | 38.5 | 43 | 31.75 |
| Baichuan-13B-4k | 0.07 | 17.55 | 17.28 | 38.55 | 18.36 |
| ALiBi-7B-4k | 0.04 | 8.13 | 17.87 | 2.89 | 7.23 |
| CLEX-7B-4k | 18.05 | 23.68 | 44.62 | 31.15 | 29.38 |

Table 3: Single-document question answering tasks in LongBench.

| Model | HotpotQA | 2WikiMQA | Musique | DuReader | AVG. |
|---|---|---|---|---|---|
| GPT-3.5-Turbo-16k | 51.6 | 37.7 | 26.9 | 28.7 | 36.23 |
| Llama2-7B-chat-4k | 25.4 | 32.8 | 9.4 | 5.2 | 18.2 |
| LongChat-v1.5-7B-32k | 31.5 | 20.6 | 9.7 | 19.5 | 20.33 |
| CodeLLaMA-7B-16k | 33.05 | 27.93 | 14.2 | 10.78 | 21.49 |
| XGen-7B-8k | 29.7 | 21.1 | 10.3 | 11 | 18.02 |
| InternLM-7B-8k | 28.7 | 22.8 | 9 | 11.1 | 17.9 |
| Vicuna-v1.5-7B-16k | 25.3 | 20.8 | 9.8 | 19.3 | 18.8 |
| Baichuan-13B-4k | 3.29 | 15 | 0.1 | 8.77 | 6.79 |
| ALiBi-7B-4k | 2.73 | 8 | 1.33 | 11.87 | 5.98 |
| CLEX-7B-4k | 28.44 | 19.53 | 9.15 | 23.21 | 20.08 |

Table 4: Multi-document question answering tasks in LongBench.

| Model | GovReport | QMSum | MultiNews | VCSUM | AVG. |
|---|---|---|---|---|---|
| GPT-3.5-Turbo-16k | 29.5 | 23.4 | 26.7 | 16 | 23.9 |
| Llama2-7B-chat-4k | 27.3 | 20.8 | 25.8 | 0.2 | 18.525 |
| LongChat-v1.5-7B-32k | 30.8 | 22.7 | 26.4 | 9.9 | 22.45 |
| CodeLLaMA-7B-16k | 28.43 | 24.18 | 26.84 | 0.79 | 20.06 |
| XGen-7B-8k | 27.3 | 20.5 | 26.2 | 2.2 | 19.05 |
| InternLM-7B-8k | 9.7 | 15.9 | 22.8 | 12.4 | 15.2 |
| Vicuna-v1.5-7B-16k | 27.9 | 22.8 | 27.2 | 15.1 | 23.25 |
| Baichuan-13B-4k | 6.8 | 1.71 | 23.1 | 8.09 | 9.925 |
| ALiBi-7B-4k | 5.31 | 1.64 | 19.38 | 3.25 | 7.395 |
| CLEX-7B-4k | 32.52 | 22.9 | 25.55 | 12.03 | 23.25 |

Table 5: Summarization tasks in LongBench.

| Model | TREC | TriviaQA | SAMSum | LSHT | AVG. |
|---|---|---|---|---|---|
| GPT-3.5-Turbo-16k | 68 | 91.4 | 41.7 | 29.2 | 57.575 |
| Llama2-7B-chat-4k | 61.5 | 77.8 | 40.7 | 19.8 | 49.95 |
| LongChat-v1.5-7B-32k | 63.5 | 82.3 | 34.2 | 23.2 | 50.8 |
| XGen-7B-8k | 65.5 | 77.8 | 25.3 | 20.5 | 47.275 |
| CodeLLaMA-7B-16k | 70 | 84.97 | 43.43 | 32.5 | 57.725 |
| InternLM-7B-8k | 52 | 77.8 | 21.2 | 15.2 | 41.55 |
| Vicuna-v1.5-7B-16k | 71.5 | 86.2 | 40.8 | 28.8 | 56.825 |
| Baichuan-13B-4k | 20.05 | 20.06 | 5.77 | 1 | 11.72 |
| ALiBi-7B-4k | 9.25 | 8.83 | 4.67 | 0 | 5.6875 |
| CLEX-7B-4k | 68 | 84.92 | 42.82 | 28.35 | 56.0225 |

Table 6: Few-shot learning tasks in LongBench.

| | Passage Count | PassageRetrieval-en | PassageRetrieval-zh | AVG. |
|---|---|---|---|---|
| GPT-3.5-Turbo-16k | 4.5 | 71 | 77.5 | 51 |
| Llama2-7B-chat-4k | 2.1 | 9.8 | 0.5 | 4.13 |
| LongChat-v1.5-7B-32k | 1 | 30.5 | 7.6 | 13.03 |
| CodeLLaMA-7B-16k | 2 | 13.5 | 11.25 | 8.92 |
| XGen-7B-8k | 2.1 | 8.5 | 3.5 | 4.7 |
| InternLM-7B-8k | 3 | 6 | 0.9 | 3.3 |
| Vicuna-v1.5-7B-16k | 6.5 | 4.5 | 5 | 5.33 |
| Baichuan-13B-4k | 0.06 | 0.5 | 5 | 1.85 |
| ALiBi-7B-4k | 0 | 1.27 | 0.75 | 0.67 |
| CLEX-7B-4k | 0 | 11.5 | 17.5 | 9.67 |

Table 7: Synthetic tasks in LongBench.

| | LCC | RepoBench-P | AVG |
|---|---|---|---|
| GPT-3.5-Turbo-16k | 54.7 | 53.6 | 54.15 |
| Llama2-7B-chat-4k | 52.4 | 43.8 | 48.1 |
| LongChat-v1.5-7B-32k | 53 | 55.3 | 54.15 |
| CodeLLaMA-7B-16k | 64.35 | 55.87 | 60.11 |
| XGen-7B-8k | 38.6 | 38.6 | 38.6 |
| InternLM-7B-8k | 44.1 | 28.8 | 36.45 |
| Vicuna-v1.5-7B-16k | 51 | 43.5 | 47.25 |
| Baichuan-13B-4k | 47.98 | 16.58 | 32.28 |
| ALiBi-7B-4k | 46.69 | 18.54 | 32.61 |
| CLEX-7B-4k | 59.01 | 56.87 | 57.94 |

Table 8: Code completion tasks in LongBench.

