# OpenReview forum: "CLEX: Continuous  Length Extrapolation for Large Language Models"
_ICLR.cc/2024/Conference — ICLR 2024 poster_

### Official Review · Reviewer_jLqk · 2023-10-17

**Soundness:** 3 good
**Presentation:** 3 good
**Contribution:** 2 fair
**Rating:** 6
**Confidence:** 4

**Summary:**

This paper proposes CLEX, a method that enables length extrapolation on Rotary position embedding (RoPE) by finetuning on a dataset. Prior work has found that finetuning with a position embedding scaling (PE scaling) on either the position or the frequency values of RoPE can enhance the extrapolation of a pre-trained language model. Based on the PE scaling, CLEX models the position-frequency values of RoPE with a neural ODE and the YaRN model, with an aim to learn the continuous dynamics over the length scaling factor during PE scaling. The experiments demonstrate a strong extrapolation compared with prior PE scaling methods (Position Interpolation, YaRN, CodeLLaMA) and non-PE scaling methods (ALiBi, RandomPos).

**Post Rebuttal Update**

I acknowledge the authors' efforts in addressing the questions. The new experiments have effectively addressed my concerns. As a result, I have raised my score.

**Strengths:**

* CLEX demonstrates strong length extrapolation results (evaluation length > train length) compared with prior PE scaling methods such as Position Interpolation,YaRN, and CodeLLaMA.

* When the evaluation length is smaller than the train length (i.e. finetuning length), CLEX exhibits a better performance compared with prior PE scaling methods.

* The ablation includes a few useful topics such as continuous vs. discrete dynamics, sampling strategy, and log-scaling.

**Weaknesses:**

* Despite a strong extrapolation performance, the motivation for adopting continuous modeling is a bit unclear. It seems that the continuous model has to be somehow discretized on a few points (e.g. evaluating the integral of equation 12). If this is true, doesn't this imply an equivalent discrete modeling?
* CLEX is adopting YaRN in equation 13, so it seems some part of the performance of CLEX is due to YaRN. An ablation of CLEX without YaRN is needed.
* CLEX is based on PE scaling, which requires a finetuning dataset. However, non-PE scaling methods (e.g., ALiBi and RandomPos) don't require finetuning. So it doesn't seem fair to compare CLEX with non-PE scaling methods.
* The author mentioned that CLEX is computationally demanding due to the evaluation on the integral. Maybe the author can comment more on the training time.
* The author claimed that AliBi-like methods (attention biasing) struggle in practical tasks requiring long-context dependency; however, the cited evidence is on AliBi. Among the author-cited AliBi-like methods, there are attention-biasing methods that achieve better long-context dependency than Alibi. Maybe the author can clarify on this.
* The notations are confusing sometimes. For example, $\lambda$ is supposed to be an amplification factor but is missing in equation 13.

**Questions:**

* In Table 1, the authors provided numbers for CLEX with training length 4k, 8k, and 16k. However, most of the other PE scaling methods (PI and YaRN) are trained only on 16k. I wonder how PI and YaRN perform when finetuning with 4k and 8k context length.

* For other questions, see the Weakness section.

---

> ### Author Response · Authors · 2023-11-22
> **Response to Reviewer jLqk (Part 1/2)**
>
> Dear reviewer, we have conducted extensive experiments, which take around 1000 A100 GPU hours, to prepare thorough responses to the raised concerns and questions. But first, we would mention that we are very shocked that while we were preparing the response, you lowered your rating score from 5 to 3 before we submitted it. Hope the responses below can ease your concerns and we look forward to your re-evaluation regarding our submission.
>
>
>
> > Q1: In Table 1, the authors provided numbers for CLEX with training lengths 4k, 8k, and 16k. However, most of the other PE scaling methods (PI and YaRN) are trained only on 16k. I wonder how PI and YaRN perform when finetuning with 4k and 8k context lengths.
>
> A1: Since the naive context length of LLaMA-2 is 4k, fine-tuning the model with PI or Yarn on 4k length is equivalent to straightforward fine-tuning, which can solely perform well within 4k resembling the native LLaMA-2 in Table 1. We have also conducted the experiments for PI and YaRN on 8k length previously, where the PPL results are shown in the table below.
>
> |     | 4k  | 8k  | 16k | 32k |
> | --- | --- | --- | --- | --- |
> | PI-8k | 5.96 | 5.76 | 6.12 | 9.34 |
> | YaRN-8k (t=4) | 6.01 | 5.84 | 6.04 | 8.84 |
> | CLEX-8k | 5.98 | 5.78 | 5.71 | 5.99 |
>
> The observations on 8k are consistent with those on 16k, hence we report the results of PI and YaRN on 16k for a representative comparison.
>
> > Q2: Despite a strong extrapolation performance, the motivation for adopting continuous modeling is a bit unclear. It seems that the continuous model has to be somehow discretized on a few points (e.g. evaluating the integral of equation 12). If this is true, doesn't this imply an equivalent discrete modeling?
>
> A2: In fact, we learn a continuous scaling function through the ODE in Eq. (12) — where we can acquire frequency bases corresponding to any t values (varying as a continuous variable). While for the inference, there is a certain sequence length and we must acquire a certain frequency basis for it. So it is necessary to sample a discrete point from the continuous function learned during training. In other words, we do learn continuous PE scaling, and the way to apply the continuous modelling for certain sequences during inference is by sampling some discrete points from the learned continuous function during training. While the discretization strategy described in Section 3.3 is used to avoid sample discrete points for each sequence, instead utilizing some predefined points to approximately apply for all evaluation sequences to speed up inference and we found it would not significantly affect the performance.
>
> To further address your concern, we include an experiment where the same two-layer NN in Eq. (13) is directly used to model the z(t) and we have z(t) = z(1) + NN($p_t$) and $p_t$ is learnable embedding for $t \in \{1, 2, 3, …, t_{max}\}$ to identify the scaling factor at discrete $t$ values. The model undergoes the same training procedure of our method as a discrete baseline. The results are listed in the table.
>
> |     | 4k  | 8k  | 16k |
> | --- | --- | --- | --- |
> | Discrete | 7.13 | 7.19 | 8.38 |
> | Ours (Continuous) | 5.86 | 5.70 | 5.87 |
>
> We found the discrete baseline performs significantly worse than our method, which demonstrates the effectiveness of our continuous modelling via the Neural ODE.
>
>
> > Q3: CLEX is adopting YaRN in equation 13, so it seems some part of the performance of CLEX is due to YaRN. An ablation of CLEX without YaRN is needed.
>
> A3: Thank you for your suggestion. We further conducted a group of experiments by removing the $\xi_t$ term in Eq. (13) and training a LLaMA-2-7B model with CLEX on  the sequences of 4K length. The results (PPL scores) of this ablation study are presented in the table below:
>
> |     | 4k  | 8k  | 16k |
> | --- | --- | --- | --- |
> | w/ $\xi_t$ | 5.86 | 5.70 | 5.87 |
> | w/o $\xi_t$ | 5.88 | 5.74 | 5.84 |
>
> From the table, we can see that CLEX with $\xi_t$ and CLEX without $\xi_t$ perform similarly across different evaluation lengths. This suggests that the $\xi_t$ term in Eq. (13) does not lead to better results of CLEX and it may not be necessary during training.
>
> However, we still recommend using the $\xi_t$ term, especially when training data is limited, as we observed that training with $\xi_t$ term converges faster (although not better), please kindly refer to Figure 10 in Appendix for training loss curves.

---

> ### Author Response · Authors · 2023-11-22
> **Response to Reviewer jLqk (Part 2/2)**
>
> > Q4: CLEX is based on PE scaling, which requires a finetuning dataset. However, non-PE scaling methods (e.g., ALiBi and RandomPos) don't require finetuning. So it doesn't seem fair to compare CLEX with non-PE scaling methods.
>
> A4: We would like to clarify that we train all models, including PE scaling and non-PE scaling ones, under the same group of experimental setups (data size, batch size, epoch, etc.) for fair comparison. The baseline models and our CLEX are shown to be convergent in terms of training loss curves (refer to Figure 9 in Appendix). Note that non-PE scaling methods such as ALiBi and RandomPos still require further training (e.g., continual pre-training or fine-tuning) as they proposed in original papers[1][2]. They cannot be applied to LLaMA (or other LLMs that are not jointly pretrained with them) in a training-free manner.
>
> [1] Train Short, Test Long: Attention with Linear Biases Enables Input Length Extrapolation
> [2] Randomized Positional Encodings Boost Length Generalization of Transformers
>
> > Q5: The author mentioned that CLEX is computationally demanding due to the evaluation on the integral. Maybe the author can comment more on the training time.
>
> A5: We wish to explain that the computationally demanding approach we mentioned in the paper refers to the implementation of acquiring each frequency basis for each sequence, which would slow the autoregressive generation. However,  our CLEX adopts the frequency cache to circumvent such issue, enabling similar inference speed (27.8 tokens/s and 28.3 tokens/s of CLEX and native LLaMA-7B, respectively on a single A100).  In terms of training time where there is no iterative passes, the computational cost of the integral involved in CLEX, which is performed using a two-layer FFN with only slight parameters, is negligible compared to the computational cost of the forward and backward passes of the LLaMA model itself.
>
> To provide a quantitative comparison, we measured the throughputs of naive FT and CLEX trained on LLaMA-2-7B with a 16k sequence length on 8xA100 GPUs. The throughputs for both methods were comparable, with values of 20,936 tokens/s for naive FT and 20,819 tokens/s for CLEX training. In other words, it would take 26.5 and 26.7 hours for the training of naive FT and our CLEX on the LLaMA-2-7B with 2B tokens, respectively. This suggests that the additional computational cost introduced by CLEX is minimal and does not significantly affect the training speed.
>
> > Q6: The author claimed that AliBi-like methods (attention biasing) struggle in practical tasks; however, the cited evidence is on AliBi. Among the author-cited AliBi-like methods, there are attention-biasing methods that achieve better long-context dependency than Alibi.
>
> A6: We now include an additional attention-biasing method, Sandwich[1], which claims to be a better attention-biasing method than ALiBi. We train the Sandwich on LLaMA-2-7B with a 4k sequence length, which undergoes the same training procedure (continual pretraining on a subset of Redpajama-Book and instruction tuning on UltraChat) as our method and ALiBi. The results of Sandwich on LongBench (maximum 16k evaluation length) are shown in the Table below.
>
> | Method | Train Length | Avg. | Single-Document QA | Multi-Document QA | Summarization | Few-shot Learning | Sythetic Task | Code Completion |
> | --- | --- | --- | --- | --- | --- | --- | --- | --- |
> | ALiBi-7B-4K | 4k  | 9.93 | 7.23 | 5.98 | 7.4 | 5.69 | 0.67 | 32.61 |
> | Sandwich-7B-4k | 4k  | 10.46 | 10.15 | 6.44 | 9.36 | 4.88 | 0.67 | 31.24 |
> | CLEX-7B-Chat-4K | 4k  | 32.72 | 29.38 | 20.08 | 23.25 | 56.02 | 9.67 | 57.94 |
>
> We can observe that Sandwich indeed performs slightly better than ALiBi, however, they both far fall behind our CLEX and cannot showcase meaningful extrapolation ability when evaluated on the LongBench. In addition, Giraffe[2] also demonstrated that xPos (which can be considered an attention-biasing method with multiplied bias) performs poorly in long-context recall tasks.  We guess that the poor performance in long-context applications may be an inherent drawback of attention-biasing methods. These methods tend to give larger biases towards nearby positions, resulting in overly relying on local context for prediction. While this may yield exceptional perplexity results (where long-term dependencies may not be crucial), it may hinder explicit access to long context, thus leading to poor performance in long-context applications.
>
> [1] Dissecting Transformer Length Extrapolation via the Lens of Receptive Field Analysis
> [2] Giraffe: Adventures in Expanding Context Lengths in LLMs
>
>
> > Q7: The notations are confusing sometimes. For example, $\lambda$ is supposed to be an amplification factor but is missing in equation 13.
>
> A7: We apologize for any confusion. The amplification factor is used to control the parameter size of $W_{down}$ and $W_{up}$ and it appears in the superscript. We will refine this notation for clearer presentation.

---

> ### Author Response · Authors · 2023-11-23
>
> Dear Reviewer jLqk,
>
>
> As we noticed that you updated your review and rating score before our response was submitted, we are more than eager to have you involved in discussions after we thoroughly replied to your questions and concerns in our response.
>
>
> Regards,
>
> Authors

---

### Official Review · Reviewer_pxa3 · 2023-11-01

**Soundness:** 3 good
**Presentation:** 3 good
**Contribution:** 3 good
**Rating:** 6
**Confidence:** 2

**Summary:**

The paper introduces CLEX, a method to efficiently extend the context window of LLMs without compromising performance. Traditional methods either have length limitations or suffer performance drops. CLEX overcomes these by modeling the relation of sequence length and frequency in RoPE during position extrapolation as a continuous system. Specifically, it utilizes Neural ODE as a tool to do this. In tests, CLEX extends context windows to over 4x the training length without performance loss and beat several popular methods both in long sequence modeling and downstream tasks.

**Strengths:**

1: The theory part is closely combined with practical part. And the performance of the proposed method also aligns with the theoretical derivation.
2: The performance of the proposed method is good. And the experiments are comprehensive.
3: Besides the main results, this paper also provided some insightful observations about LLMs' length generalization.

**Weaknesses:**

Please check the question section

**Questions:**

1: By my understanding, the core idea of this paper is to do position extrapolation with an appropriate frequency for different sequence lengths. Is it necessary to utilize Neural ODE? In another word, can we use a regular NN? As it seems that a regular NN can do the same thing. (Not quite sure, for I'm not an expert of Neural ODE)

2: Could you please provide more details about the training/fine-tuning? Did you train all the baseline models with the same number of tokens, the same batch size as well as the same steps? If so, for PI, its paper mentioned that they only fine-tuned the LLM for ~ 1000 steps, while for some other baselines such as replacing RoPE with Alibi for Llama-2, the tuning steps should definitely be much larger. With different required number of training steps is the performance comparison fair enough?

3: Also, I'm wondering if there's any explanation to the poor performance on LongBench's synthetic tasks.

4: Compared to Random Position, the main difference is that CLEX add adaptive frequency for different sequences, it that correct?

5: Still about Random Position, its original paper and some blogs (https://kexue.fm/archives/9444, it's in Chinese, you may translate it to English first) showed that it shows good length generalization ability. But in  Table-1, Random Position does not work at all ( trained on 4k, and can only keep low PPL at 4k), do you have any thoughts about it?

---

> ### Author Response · Authors · 2023-11-22
> **Response to Reviewer pxa3 (Part 1/2)**
>
> We sincerely thank the reviewer for the insightful comments and detailed feedback. Please kindly find our response below.
>
> > Q1: By my understanding, the core idea of this paper is to do position extrapolation with an appropriate frequency for different sequence lengths. Is it necessary to utilize Neural ODE? In another word, can we use a regular NN? As it seems that a regular NN can do the same thing. (Not quite sure, for I'm not an expert of Neural ODE)
>
> A1: We provide a unified view of PE scaling: an ODE based on continuous dynamic in Eq. (11). Our network directly models the dynamic ($g_{\phi}$) here, to form a standard Neural ODE. This unified view can be also interpreted as some special cases. For example, let us consider using the same two-layer NN in Eq. (13) to model the $z(t)$ straightforward, where we have $z(t) = z(1) + NN(p_t)$ and $p_t$ is learnable embedding for $t \in [1, t_{max}]$ to identify the scaling factor at $t$ value. We train a model using the regular NN that undergoes the same training procedure of our method as a discrete baseline. The results are listed in the Table below:
>
> |     | 4k  | 8k  | 16k |
> | --- | --- | --- | --- |
> | Discrete (Regular NN) | 7.13 | 7.19 | 8.38 |
> | Ours (Continuous ODE) | 5.86 | 5.70 | 5.87 |
>
> We found the discrete baseline performs significantly worse than our method, which demonstrates the effectiveness of our Neural ODE.
>
> > Q2: Could you please provide more details about the training/fine-tuning? Did you train all the baseline models with the same number of tokens, the same batch size as well as the same steps? If so, for PI, its paper mentioned that they only fine-tuned the LLM for ~ 1000 steps, while for some other baselines such as replacing RoPE with Alibi for Llama-2, the tuning steps should definitely be much larger. With different required number of training steps is the performance comparison fair enough?
>
> A2: Yes, we train all baseline models using the same subset of 2B tokens from Redpajama-Book. We deem that 2B tokens are enough for all methods to converge, including PI and ALiBi (also they both propose that they require minimal training steps). If not, the method is probably over-expensive to utilize. We use FLASHATTENTION and FSDP for training. The batch size is set as 64k tokens per device (global batch size of 512k tokens on 8x A100 GPUs) for all PE scaling methods, naive FT, and RandomPos. We implement ALiBi with the Triton version of FLASHATTENTION (since the CUDA version does not support attention bias input), which takes more GPU memory than the CUDA version. So the batch size of ALiBi is 32k tokens per device, which means ALiBi takes twice the training steps (but total training tokens are the same).  In Figure 9 in the appendix, we found all methods converge well, so we deem that we have dug out the full performance of all methods in our experiments, as well as that the comparison is fair and should be convincing.
>
> > Q3: Also, I'm wondering if there's any explanation to the poor performance on LongBench's synthetic tasks
>
> A3:  The synthetic tasks in LongBench include two types: passage-count and passage-retrieval. Passage-count requires models to count the number of paragraphs in a passage while passage-retrieval requires models to identify a specific paragraph to which a given summarization corresponds. Both tasks need models to output the numbers, however, LLaMA-based models may be not good at counting and calculating without CoT or other prompt engineering methods, as well as the output number may be unstable and easy to change. We think this may be the reason why all LLaMA-based models including ours showcase poor performance on synthetic tasks. This may be improved by carefully designing the prompt (e.g., CoT), but here we keep the native setting in LongBench for a fair comparison with all baselines.

---

> ### Author Response · Authors · 2023-11-22
> **Response to Reviewer pxa3 (Part 2/2)**
>
> > Q4: Compared to Random Position, the main difference is that CLEX adds adaptive frequency for different sequences, it that correct?
>
> A4: We would like to clarify that there are distinct differences between Random Position and CLEX. Actually, the random position is not necessary in CLEX. As shown in Figure 4 (middle), CLEX with a random position or uniform position (sampling position ids from $[1, tL]$ at a fixed distance $t$) achieves similar performance. This demonstrates the effectiveness of our position extrapolation strategy for addressing the inconsistency between frequency basis and position indices (as we discussed in Sec. 3.3, paragraph 2), whatever we use random sampling or uniform sampling to implement it.
>
> > Q5: Still about Random Position, its original paper and some blogs (https://kexue.fm/archives/9444, it's in Chinese, you may translate it to English first) showed that it shows good length generalization ability. But in Table-1, Random Position does not work at all ( trained on 4k, and can only keep low PPL at 4k), do you have any thoughts about it?
>
> A5: We found that another work (https://arxiv.org/abs/2309.10400, Table 1) reported similar evaluation results to ours, where Random Position can only achieve good PPL scores within training length (or even less than training length).
>
> We think there are two probable reasons: first, the results in the original paper (shown in the blog you mentioned) are evaluated on algorithmic reasoning tasks rather than natural language tasks, and the performance of RandomPos on natural language modelling remains questionable.
>
> Second, the original experiments of RandomPos are conducted on a naive Transformer rather than a pretrained LLM. For educated guess, there may be an issue that each position index in LLMs is “pretrained” many times on a quite large corpus, whereas the randomly sampled position indices are just fine-tuned with fewer frequencies. The LLM pretraining may make the model fit well on original position indices, but difficult to generalize to larger ones with Random Position. While if we seek to make the random positions achieve the frequency of pretrained position indices, we may need to use corpus times larger than the pretrained one, as each sequence is equipped with sparse position indices when using Random Position. We think it is over-expensive while our method may provide a more efficient solution.

---

### Official Review · Reviewer_nREr · 2023-11-01

**Soundness:** 3 good
**Presentation:** 3 good
**Contribution:** 3 good
**Rating:** 8
**Confidence:** 2

**Summary:**

The paper proposes a new positional embedding scaling to be used for using a model with different context lengths than seen during training.
The idea is an extension of rotary positional embedding, for which the frequencies used are dynamically updated depending on the desired context length. The method used to actually update those frequencies is through a neural ODE whose parameters are also trained.

**Strengths:**

I think the proposed method has some value. It sticks to a well established PE scheme, and then proposes a way to update its parameters that is _trained_ to be good, instead of just wishing it will be based on some assumptions. For this reason, the paper definitely deserves consideration in my opinion.

The proposed method moreover seems to be providing good performance for extrapolation, which was the intent.

**Weaknesses:**

* Not much details is provided in the main text regarding how we train such a beast. I must say this looks quite daunting to me how I would train a NODE along my transformer model. I guess it would help to have some explanations to it.
* I am missing some exploration of what the model is producing regarding the frequencies for ROPE. As I understand, it boils down to being able to produce a new set of frequencies for ROPE to use for any input lengths. This would have been feasible to actually display that. Since many people have played with the idea of manually setting such parameters, I am curious whether a trained method could give us insights as to what good frequencies actually look like. Are we observing high frequencies to disappear to favor long term dependencies? Such things.

**Questions:**

The paper is mostly interesting and I guess that anyone working on the topic would have a few questions
* the random sampling method you propose look like a strong and nice ingredient of your approach. Could you just make it clear for me whether the _order_ of the samples is maintained within the sequence?
* Your method definitely allows some extrapolation as per your experiments. However, I somehow feel that it could also shine for "superresolution"/"interpolation", i.e. infilling missing data within a sequence. This feeling comes from your random sampling idea. It looks like you are basically simulating "missing data".
* "Unlike the previous PE scaling methods built on a larger scaling factor would lead to inferior performance on the lengths corresponding to smaller counterparts, the continuous PE scaling would enable non-destructively generalisation to larger scaling factors via adaptive continuous dynamics". This would be great, but at this point in the paper, I don’t see why the proposed scaling method would _necessarily_ enable it. Maybe you could rephrase that in a more humble way

---

> ### Author Response · Authors · 2023-11-22
> **Response to Reviewer nREr (Part 1/2)**
>
> We sincerely thank the reviewer for the positive review and insightful feedback. Please kindly find our response below.
>
> > Weakness 1: Not much details is provided in the main text regarding how we train such a beast. I must say this looks quite daunting to me how I would train a NODE along my transformer model. I guess it would help to have some explanations to it.
>
> Explanation 1: We apologize for any omission of the training details. To train a transformer model using our method, the only thing need to do is acquiring the frequency basis before the forward of transformer layers, subsequently, we just pass the frequency basis to all layers as the basis of RoPE. In detail for acquiring the frequency basis, we randomly sample $t \in [1, t_{max}]$ for each training batch where the $t_{max}$ is predefined (as 16 in our experiments), and then acquire the $z(t)$ through the integral in Eq. (12), where the ODE is modelled by a two-layer FFN in Eq. (13) and the integral is solved by torchdiffeq (https://github.com/rtqichen/torchdiffeq). Finally, we apply the position extrapolation to enlarge the position ids $\{1, 2, .., L\}$ to $[1, tL]$ by random sampling or uniform sampling as discussed in Sec. 3.3 to calculate the cos and sin embeddings in RoPE.
>
> We use FLASHATTENTION-V2, gradient checkpointing, and FSDP for all models’ training. With BFloat16 mixed precision, it would need 2x and 4x A100 80G GPUs for training 7B-sized models on 4k and 16k context lengths, respectively. (But we recommend using twice the number of GPUs for larger batch sizes and faster training.) To train a 13B-sized model over 4k length, it may be necessary to use 8xA100-80G GPUs or apply weight quantization.
>
> We hope these details can help you understand how to train a custom transformer model using our method. We will release all of the codes, model checkpoints, and training scripts when the paper is publicly available.
>
> > Weakness 2: I am missing some exploration of what the model is producing regarding the frequencies for ROPE. As I understand, it boils down to being able to produce a new set of frequencies for ROPE to use for any input lengths. This would have been feasible to actually display that. Since many people have played with the idea of manually setting such parameters, I am curious whether a trained method could give us insights as to what good frequencies actually look like. Are we observing high frequencies to disappear to favor long term dependencies? Such things.
>
> Explanation 2: Thank you for your insightful comment. We additionally plot the frequency bases learned by CLEX over the dimensions, compared to the original basis with a base of 10,000 and the enlarged base of 1,000,000 in CodeLLaMA (please kindly refer to **Figure 8** in the revised submission). The plot reveals that CLEX tends to upscale the high frequencies at certain dimensions while simultaneously downscaling some others. More surprisingly, we have observed that the frequency bases associated with different t values in CLEX exhibit an isotropic behaviour, that the dimensions where downscaling and upscaling occur are similar across different t values, with larger t values resulting in further scaling.

---

> ### Author Response · Authors · 2023-11-22
> **Response to Reviewer nREr (Part 2/2)**
>
> > Q1: The random sampling method you propose look like a strong and nice ingredient of your approach. Could you just make it clear for me whether the *order* of the samples is maintained within the sequence?
>
> A1: We apologize for the confusion about the random sampling method. The underlying idea of random sampling is to learn explicit associations between larger position indices and the learned frequency basis corresponding to the scaling factor at the value of sampled t (as discussed in paragraph #2 of Sec. 3.3). Let us illustrate by an example for a training sequence with 4096 length: for a specific training step, we would sample a $t \in [1, t_{max}]$ and acquire the frequency basis at scaling factor $t$ by the ODE in Eq. (13). That means the acquired frequency basis corresponds to a sequence with a length of $4096*t$. While in the original RoPE (see Eq. (1)), the position indices ($m$) are bounded with the frequency basis ($\theta$). However, in the current training sequence, the position indices range in $[1, 4096]$, resulting in the inconsistency between position indices and frequency basis. So we are necessary to enlarge the position indices from the scope of  $[1, 4096]$ to that of $[1, 4096t]$ to address the inconsistency. This can be implemented by randomly sampling 4096 indices in **ascending order** from $[1, 4096t]$ (e.g., ${1, 3, 8, 11, 19, …}$), or uniform sampling as $\{t, 2t, 3t, …, 4096t\}$. In our ablation in Figure 4 (middle), uniform sampling works well while random sampling performs slightly better than uniform sampling.
>
> > Q2: Your method definitely allows some extrapolation as per your experiments. However, I somehow feel that it could also shine for "superresolution"/"interpolation", i.e. infilling missing data within a sequence. This feeling comes from your random sampling idea. It looks like you are basically simulating "missing data".
>
> A2: Thank you for your feedback, which is a very interesting explanation! Our CLEX indeed simulates the "missing frequency bases" by the ODE  in Eq. (13). While the position extrapolation strategy by random sampling resembles approaching the “missing position indices” (i.e., indices beyond the original training length) to mitigate the gap to the predicted frequency bases. The reason why we call it “extrapolation” is because we use a small scope (e.g., position indices in [1, L]) to simulate a larger scope (e.g., position indices in [1, tL]). Likewise, there is potential for applying a similar approach in reverse to shrink a scope and predict the missing data within this smaller scope. This may be helpful for the image domain (e.g., superresolution) and potential to be used in Autoregressive image encoder/decoder.
>
> > Q3: "Unlike the previous PE scaling methods built on a larger scaling factor would lead to inferior performance on the lengths corresponding to smaller counterparts, the continuous PE scaling would enable non-destructively generalisation to larger scaling factors via adaptive continuous dynamics". This would be great, but at this point in the paper, I don’t see why the proposed scaling method would *necessarily* enable it.
>
> A3: According to our unified view in Theorem 1 and Eq. (8), the working mechanisms of previous PE scaling methods are similar, i.e., perform manipulation over the frequency basis. They usually involve fine-tuning on a pre-set and fixed frequency basis, which makes the context length efficiently extend to the desired length. If we pre-train with this frequency basis, it definitely performs well over all lengths within the training sequence length. But now the fact is that we first pre-train the model on a huge corpus (trillion level) using the original frequency basis, yet seek to fine-tune it using another frequency basis on a much smaller corpus through minimum training steps as they proposed. As shown in Figure 1, these methods (e.g. PI and Yarn) are slightly worse than naive fine-tuning using the original frequency basis, on short lengths within the pre-training length (4k). (This performance drop within short lengths is also mentioned in CodeLLaMA.) Although the performance drops are not quite significant, this would make us unable to confidently apply them to longer lengths.
>
> In contrast, our method makes the model adapt to different frequency bases corresponding to different t values at different training steps, avoiding overfitting specific frequency bases. More essentially, these frequency bases are modelled by a Neural ODE in Eq. (12) with a two-layer FFN as parameters. This provides a fine-grained optimization, where intuitively the gradient would be harnessed towards improving the performance for every frequency basis. So when the loss comes to converge, the model should adapt to frequency bases corresponding to a continuous scope of the scaling factor. During inference, we could apply the learned frequency bases correspondingly for sequences of various lengths without performance drops.

---

### Official Review · Reviewer_ZBA8 · 2023-11-01

**Soundness:** 3 good
**Presentation:** 3 good
**Contribution:** 2 fair
**Rating:** 6
**Confidence:** 5

**Summary:**

The paper studies the length extrapolation problem of large language models, i.e., training on short sequences while testing on long sequences. The work is built upon RoPE. Continuous PE scaling is introduced as a RoPE embedding scaling method.

**Strengths:**

Originality:
Continuous PE scaling is introduced as a RoPE embedding scaling method.


Clarity:
The paper is easy to follow and understand.

Significance:
Long-sequence modeling is important for many downstream applications.

**Weaknesses:**

- The work is built upon RoPE, which limits its application to other models that don't use RoPE.

- According to Table 1, the models still do not perform "real" length extrapolation. The PPL results become worse when the length is increased. If PPL becomes worse, why not directly use window-based methods in practice? The real-world value of the proposed method is questionable.

- Straightforward method (such as https://arxiv.org/abs/2309.16039) works well in practice. It also challenges the value of research on length extrapolation, as long as we finetune the models. So the evaluation setting can be improved.

- Fig 5 indicates that different models perform similarly across tasks, despite GPT. The significance of the method is not clearly demonstrated.

**Questions:**

- The work is built upon RoPE, which limits its application to other models that don't use RoPE. How to use the proposed method for other PE methods?

- According to Table 1, the models still do not perform "real" length extrapolation. The PPL results become worse when the length is increased. If PPL becomes worse, why not directly use window-based methods in practice? The real-world value of the proposed method is questionable.

- Straightforward method (such as https://arxiv.org/abs/2309.16039) works well in practice. It also challenges the value of research on length extrapolation, as long as we finetune the models. The proposed method can be integrated into the above pipeline, which provides more valuable evaluation metrics.

- Fig 5 indicates that different models perform similarly across tasks, despite GPT. The significance of the method is not clearly demonstrated.

- The lines in the right subfigure of Fig 5 are not correctly shown. The figure can be updated.

---

> ### Author Response · Authors · 2023-11-22
> **Response to Reviewer ZBA8 (Part 1/2)**
>
> We sincerely thank the reviewer for the insightful comments and detailed feedback. Please kindly find our response below.
>
> > Q1: The work is built upon RoPE, which limits its application to other models that don't use RoPE.
>
> A1: Thank you for your comments. Our method is indeed designated for RoPE since encoding positions with RoPE has become a common practice in modern LLM architectures (e.g., LLaMA, Falcon, Mistral, …). On the other hand, we would like to clarify that our method only requires the manipulation over frequency basis and it can be applied to arbitrary position encoding methods that rely on frequency input (e.g., sinusoidal position embeddings).
>
> > Q2: According to Table 1, the models still do not perform "real" length extrapolation. The PPL results become worse when the length is increased. If PPL becomes worse, why not directly use window-based methods in practice? The real-world value of the proposed method is questionable.
>
> A2: We would like to clarify that the term “length extrapolation” in our paper refers to the capability of training on short texts while testing on longer ones. Although our method does not support the input of infinite length yet, it is still capable of expanding the context window size of LLMs to 4~8x of training length with almost NO performance drop, as shown in Table 1. Taking CLEX-16K as an example, given a training length of just 16K, our method enables LLaMA-2 to effectively handle the input of up to 64K tokens, which largely meets the requirement of several long-context applications, such as multi-document QA and text summarization, at an affordable cost.
>
> For the window-based methods, despite their support of infinite input length, they may suffer from the issues of information loss and context fragmentation when dealing with very long sequences. We conducted a pilot experiment using StreamingLLM, which is a recent window-based method, on CodeLLaMA-7B with a window size of 4000 and a number of attention sink tokens of 64. The table below shows the performance of StreamingLLM on LongBench.
>
> | Method | Avg. | Single-Document QA | Multi-Document QA | Summarization | Few-shot Learning | Synthetic Task | Code Completion |
> | --- | --- | --- | --- | --- | --- | --- | --- |
> | CodeLLaMA-7B-16k | 33.42 | 32.19 | 21.49 | 20.06 | 57.73 | 8.92 | 60.11 |
> | CodeLLaMA-7B-16k (w/ StreamingLLM) | 11.12 | 5.22 | 1.40 | 10.92 | 14.02 | 0   | 35.15 |
> | CLEX-7B-4k | 32.72 | 29.38 | 20.08 | 23.25 | 56.02 | 9.67 | 57.94 |
>
> The results indicate that the performance of CodeLLaMA on LongBench tasks significantly dropped when limited to a window of 4000 tokens using StreamingLLM compared to full attention over the maximum 16k context length. This suggests that window-based methods may struggle to capture the full information and context history, leading to compromised performance.
>
> In contrast, our method, trained on 4k tokens, achieves comparable performance to CodeLLaMA trained on 16k tokens. This demonstrates that our method effectively extends the context length capabilities of full attention models. While there may be an upper bound to the extrapolation scope, the exceptional extrapolation ability we achieve (even over 64k) demonstrates the effectiveness of our method for full attention in real-world applications, particularly in scenarios that require accurate and comprehensive context understanding.

---

> > ### Author Response · Authors · 2023-11-22
> > **Response to Reviewer ZBA8 (Part 2/2)**
> >
> > > Q3: Straightforward method (such as https://arxiv.org/abs/2309.16039) works well in practice. It also challenges the value of research on length extrapolation, as long as we finetune the models. So the evaluation setting can be improved.
> >
> > A3: Thanks for your feedback. We agree that the mentioned straightforward method, i.e., long-context fine-tuning, is simple yet effective. However, such a method is computationally unaffordable when scaling to very large input lengths.  For instance, training a 16k-context 7B model via long-context fine-tuning on 16k length would require 8xA100 80G GPUs (batch size=2) (with FLASHATTENTION, bfloat16, gradient checkpointing, FSDP/Deepspeed, etc.). Besides, when the fine-tuning length reaches 24k for 7B model (hidden size=4096), the computation latency of attention over the long sequence during training would be a heavy bottleneck[1]. In contrast, our CLEX method allows us to finetune on shorter lengths (e.g. 4k) but it is able to extrapolate to 16k, requiring less computational resources(e.g. 2xA100 80G GPUs over batch size=2) and speeding up the training. This significant reduction in GPU consumption makes our method more efficient and cost-effective.
> >
> > [1] Efficient Large-Scale Language Model Training on GPU Clusters using Megatron-LM
> >
> > > Q4: Fig 5 indicates that different models perform similarly across tasks, despite GPT. The significance of the method is not clearly demonstrated.
> >
> > A4: We apologize for the misleading and confusion. In Fig 5, we do not seek to either claim SOTA on certain datasets or demonstrate superiority to certain baselines. Instead, we want to validate the effectiveness of our CLEX on real tasks (other than language modelling) and show how close it can be to the compared methods trained on longer sequences. Specifically, our CLEX-7B-4k, which is solely trained on sequences of 4K length, performs similarly to LongChat-v1.5-7B-32k, CodeLLaMA-7B-16k, and Vicuna-v1.5-7B-16k (trained on sequences of 32K, 16K, and 16K lengths respectively) across tasks, suggesting that the length extrapolation strategy of our CLEX works well on not only language modelling but also real NLP tasks.
> >
> > > Q5: The lines in the right subfigure of Fig 5 are not correctly shown. The figure can be updated.
> >
> > A5: We apologize for any confusion created by the right subfigure of Fig 5. This scatter showcases the average performance of models across all tasks in the LongBench, associated with the training sequence length. The dotted line denotes the performance of our method. This figure shows normally in many different pdf readers (e.g. Preview and Chrome), but please let us know if there is any pdf reader where this figure could not plot properly.This figure aims to clarify that our method trained on 4k only achieves comparable performance with all open-source models trained on lengths up to 32k.

---

### Author Response · Authors · 2023-11-23
**General Response to All Reviewers and ACs**

We thank the reviewers for their insightful comments and constructive feedback. We would like to highlight that we propose a unified theoretical framework for position embedding (PE) scaling and utilize a lightweight yet effective neural ODE to model it. This allows for the extrapolation of the context length of LLMs to more than 4x training length. For instance, an LLM trained on sequences of 16k length would be able to process the inputs of up to 64k tokens without any performance drop. Our method is designated for RoPE, considering that RoPE is the most prevalent position encoding method in modern LLMs (e.g., LLaMA, Falcon, and Mistral), we believe our CLEX would be of considerable benefit to the open-source LLM community.


We further conducted several experiments to address some concerns and questions from the reviewers, along with supportive analyses. These additional experiments took around 986 A100 GPU hours, so we could not manage to post our responses earlier. To be specific, we summarize the additional experiments and analyses as follows.

1. We include the comparison among window-based CodeLLaMA, which is implemented using the method from StreamingLLM, naive CodeLLaMA and our method. It takes around 16 A100 GPU hours for evaluation. (Comment from reviewer ZBA8)

2. We add the comparison to a baseline that uses a regular NN to model our proposed PE scaling framework in a discrete manner rather than using our Neural ODE. It takes around 230 A100 GPU hours for training and evaluation. (Comments from reviewer pxa3
 and jLqk.)

3. Provide the experimental results of PI and YaRN on 8k, which we have done before but not included due to the limit of space. (Comment from reviewer jLqk)

4. Provide an ablation study for the constant dynamics used in Eq. (13), along with the training curves in Figure 10 in the Appendix to demonstrate the reason why we choose to use it (for speeding up convergence). It takes around 230 A100 GPU hours.  (Comment from reviewer jLqk)

5. Provide experimental results for Sandwich, another attention-biasing method, on the LongBench. It takes around 230 A100 GPU hours for continual pretraining and 280 GPU hours for instruction tuning. (Comment from reviewer jLqk)

6. Visualize the learned frequency basis from our method in Figure 8 of Appendix, where we found an interesting isotropic behaviour. (Comment from reviewer nREr)

7. Visualize the training curves of all baselines and our method in Figure 9, to show that all models converge well and our comparison is convincing. (Comments from reviewer pxa3 and jLqk)

These experimental results are shown in individual responses and will be incorporated into our manuscript in the next revision.  We also provide substantial discussions with all reviewers regarding training details, insights derived from our method, random position, etc. We kindly request all reviewers and ACs to carefully review our revised submission and take into account our individual responses provided below.

---

### Meta-Review · Area_Chair_nB73 · 2023-12-07

**Metareview:**

This paper introduces Continuous Length EXtrapolation (CLEX), a method that generalizes position embedding scaling approaches in large language models to effectively extend the context window beyond training sequence length without sacrificing performance, demonstrating its seamless incorporation into LLMs and competitive performance on the LongBench benchmark. There has been a consensue of the reviewers that the paper is a solid contribution to the field. I would recommend acceptance of this paper.

**Justification For Why Not Higher Score:**

n/a

**Justification For Why Not Lower Score:**

n/a

---

### Decision · Program_Chairs · 2024-01-16

Accept (poster)